# Prevalence of Undernutrition in Community-Dwelling Older Adults in The Netherlands: Application of the SNAQ^65+^ Screening Tool and GLIM Consensus Criteria

**DOI:** 10.3390/nu15183917

**Published:** 2023-09-09

**Authors:** Yaren Zügül, Caroline van Rossum, Marjolein Visser

**Affiliations:** 1Department of Health Sciences, Faculty of Science, Amsterdam Public Health Research Institute, Vrije Universiteit Amsterdam, 1081 HV Amsterdam, The Netherlands; 2National Institute for Public Health and the Environment (RIVM), 3721 BA Bilthoven, The Netherlands; caroline.van.rossum@rivm.nl

**Keywords:** aging, undernutrition, Global Leadership Initiative on Malnutrition, national survey

## Abstract

The aim of this study was to provide recent data on the prevalence of undernutrition based on screening and diagnosis in Dutch community-dwelling older adults. The data from the 2021 to 2022 examination wave from the Longitudinal Aging Study Amsterdam (*n* = 1138) and the Dutch National Food Consumption Survey 2019–2021 (*n* = 607) on community-dwelling men and women aged 65 years and older were used. The prevalence of undernutrition was based on a positive score on the Short Nutritional Assessment Questionnaire 65+ (SNAQ^65+^) screening tool, a positive diagnosis using the Global Leadership Initiative on Malnutrition (GLIM) criteria and their combination. Of the combined sample (*n* = 1745), the mean age was 74 (SD 6) years, where 16.7% were aged 80 years or older, 50.5% was female, 56.9% had a high education level, and 30.3% lived alone. The prevalence of undernutrition based on the SNAQ^65+^ screening in the combined sample was 8.5% (95% CI 7.3–9.9%). In the subgroup of LASA participants with complete data on all GLIM criteria (*n* = 700), the prevalence of undernutrition was 5.4% based on SNAQ^65+^ and 7.1% based on GLIM. A positive SNAQ^65+^ screening followed by a positive GLIM diagnosis resulted in a lower prevalence (3.1%). Being female, older, living alone, receiving formal home care, and having poor self-rated health, poor appetite, or mobility limitations, they were all associated with a higher prevalence, with more than two-fold higher prevalence rates in some subgroups. The results show that currently one out of twelve community-dwelling adults aged 65 years and older is undernourished based on the SNAQ65+ screening, and one out of fourteen is undernourished based on the GLIM diagnosis criteria. Awareness is needed to increase early recognition and treatment in community and primary care, especially among the more vulnerable groups.

## 1. Introduction

Life expectancy has increased worldwide, especially in people older than 65 years [1]. This global increase includes The Netherlands, where the proportion of adults aged 65 years and above is predicted to increase from 20% to 25% by 2040 [2]. Undernutrition is a major health concern in community-dwelling older adults [3]. The European Society for Clinical Nutrition and Metabolism (ESPEN) defines undernutrition as “a state resulting from lack of intake or uptake of nutrition that leads to altered body composition (reduction in lean mass) and body cell mass leading to diminished physical and mental function and impaired clinical outcome from disease” [4]. Older adults are particularly vulnerable to undernutrition due to natural age-related changes, such as unfavorable changes in body composition, dentition, and appetite [5,6]. Undernutrition in older adults is associated with adverse health outcomes including functional decline [7], diminished immune response [8], impaired quality of life [9,10], increased utilization of healthcare services [11], and increased mortality [12]. Thus, the achievement and maintenance of good nutritional status in older adults is critical to ensure their overall health, functioning, and quality of life [13,14].

About 10 years ago, estimates of the prevalence of undernutrition in community-dwelling older adults in The Netherlands were published based on the SNAQ^65+^ screening tool (Short Nutritional Assessment Questionnaire 65+ [15]). This tool has been developed for The Netherlands and is the most frequently used tool for older adults in the community setting in this country. The prevalence of undernutrition, defined as having a low mid-upper arm circumference (MUAC < 25 cm) and/or unintentional weight loss ≥4 kg in the past 6 months, ranged from 10.7% to 11.8% in community-dwelling older adults. A higher prevalence (34.8%) was assessed by home care nurses among older adults receiving home care [15]. The prevalence of being at risk of undernutrition, based on having a poor appetite in the previous week in combination with difficulties climbing stairs, ranged from 2.2% to 7.7%, and it was 9.2% for those receiving home care [15]. While these prevalences are lower compared to those in institutionalized older adults, in absolute numbers, the largest number of undernourished older adults live at home [16]. Unfortunately, recent prevalence data using the exact same screening tool on Dutch community-dwelling older adults, comprising 94% of all older adults in The Netherlands [17], are lacking. Using consistent methodology allows the potential comparison of prevalences over time in The Netherlands. 

In 2018, the Global Leadership Initiative on Malnutrition (GLIM) developed a set of evidence-based criteria as a framework for diagnosing undernutrition in adults [18]. According to the GLIM criteria, the diagnosis of undernutrition should be made based on the phenotypic criteria (i.e., unintentional weight loss, low body mass index (BMI), and reduced muscle mass) and the etiological criteria (i.e., reduced food intake/assimilation and inflammation). At least one phenotypic criterion and one etiological criterion are required to diagnose undernutrition. The severity of undernutrition can be based on the phenotypic criteria [18]. There are currently no studies reporting the prevalence of undernutrition in Dutch community-dwelling older adults using these GLIM criteria. A prevalence rate according to these newly developed and global criteria for the assessment of undernutrition importantly allows comparisons of prevalences between countries and between settings. Therefore, the aim of this study was to determine the current prevalence of undernutrition in Dutch community-dwelling older adults using the SNAQ^65+^ screening tool as well as the GLIM criteria for diagnosis.

## 2. Materials and Methods

### 2.1. Design

A cross-sectional observational study was performed using the recently collected data from two nationally representative samples of community-dwelling older adults in The Netherlands: the 2021–2022 examination wave of the Longitudinal Aging Study Amsterdam (LASA) and the Dutch National Food Consumption Survey 2019–2021 (DNFCS). The LASA study was approved by the Ethics Review Board of the VU University Medical Center and informed consent was obtained from every participant. Ethical review and approval were waived for the DNFCS study. The Utrecht University Medical Ethical Review Committee evaluated that the DNFCS study was not subject to the Medical Research Involving Human Subjects Act (WMO) of The Netherlands. Informed consent was obtained from all subjects involved in the DNFCS study.

### 2.2. Study Samples

LASA is an ongoing longitudinal study, initially based on a nationally representative sample of older adults aged 55–84 years from the western part of The Netherlands (in and around Amsterdam), in the northeast (in and around Zwolle), and in the south (in and around Oss). The study is focusing on the predictors and consequences of changes in autonomy and wellbeing in the aging population. A total of 3107 older adults were included at the baseline examination (1992/1993). A second cohort (2002/2003) and third cohort (2012/2013), using the same sampling frame as the original cohort, included 1002 and 1023 adults aged 55–64 years, respectively. Follow-up examinations have been performed with intervals of 3 years and consist of general and medical interviews with anthropometric measurements performed by trained interviewers in the participants’ home and a self-administered questionnaire. For the oldest respondents (80+ years) who initially did not respond and for whom filling out a questionnaire appeared to be too difficult, telephone interviews were offered. The details on the design, sampling, and data collection of LASA have been described elsewhere [19,20,21]. The data for the present study were collected in the examination wave of 2021/2022, in which 1050 older adults participated in medical interviews during which anthropometric measurements were made, and 195 adults in telephone interviews. After the exclusion of individuals younger than 65 years (*n* = 30), those with missing data on unintentional weight loss in the past 6 months (*n* = 7) and those with missing data on both BMI and MUAC (*n* = 70), the analytic sample included 1138 older adults to determine the prevalence of undernutrition based on the SNAQ^65+^ screening tool (=SNAQ^65+^ sample). To determine the prevalence of undernutrition based on the GLIM criteria, the participants with additional missing data on any of these criteria (calf circumference (*n* = 171), reduced food intake (*n* = 400), or chronic disease (*n* = 124)) were excluded, resulting in a sample of 700 participants (=GLIM sample).

DNFCS is a cross-sectional survey carried out among the general Dutch population, aged 1–79 years (*n* = 3570). The data were collected from 2019 to 2021 using a consumer panel (Kantar) in order to obtain a representative sample based on the age, sex, education, region, and urbanization level. The methodological details of the study have been described elsewhere [22]. Questionnaires were used to collect general data on the background and lifestyle factors of participants. Height and body weight were self-reported. In DNFCS, the prevalence of undernutrition was assessed using the SNAQ^65+^ screening tool only. Only the 71–79 year-olds were visited at home, during which the mid-upper arm circumference was measured, unless a home visit was not possible due to COVID-19 restrictions. For those aged 65–70 years, and for those without a home visit, the BMI was obtained. This study used the data of all 607 participants aged 65–79 years. 

### 2.3. Assessment of Undernutrition

#### 2.3.1. SNAQ^65+^ Screening 

The SNAQ^65+^ was used for nutritional screening. Older adults with a low mid-upper arm circumference (MUAC) and/or involuntary weight loss ≥4 kg in the past 6 months were defined as undernourished. Older adults who reported having a poor appetite in the past week in combination with mobility limitations were defined as being at risk of undernutrition. 

#### 2.3.2. GLIM Diagnosis

The GLIM includes three phenotypical criteria (weight loss, low BMI, and reduced muscle mass) and two etiological criteria (reduced food intake or absorption, and increased disease burden or inflammation) [18]. At least one phenotypic criterion and one etiologic criterion were required to diagnose undernutrition [18]. The severity of undernutrition was determined based on the phenotypic criteria. Undernutrition was classified as severe if the older adult had a weight loss >10% within the past 6 months, very low BMI (<18.5 kg/m^2^ for age < 70 y and <20 kg/m^2^ for age ≥ 70 y), or severe deficit in muscle mass (calf circumference ≤ 32 cm in men and ≤31 cm in women) [18,23]. Otherwise, undernutrition was classified as moderate [18].

### 2.4. SNAQ^65+^ and GLIM Criteria Used to Assess Undernutrition

#### 2.4.1. Involuntary Weight Loss

In LASA, involuntary weight loss in the past 6 months was determined using the answers for the following questions: (1) “Did your weight change in the past 6 months?”; (2) “How many kilograms did your weight change?”; and (3) “What is the main reason of your weight change?”. Involuntary weight loss was defined as weight loss from the following causes: sickness (including operation, hospitalization, not feeling well, clinical depression, or medication); social reasons (like stress, death of a spouse, retirement, or busyness); or for reasons unknown. In DNFCS, involuntary weight loss was assessed with the following question: “Did you involuntary lose 4 kg or more in the past 6 months?” (yes, no, or the participant does not know). For both LASA and DNFCS, involuntary weight loss ≥4 kg in the past 6 months was used to indicate weight loss in SNAQ^65+^, and involuntary weight loss ≥5% in the past 6 months was used to indicate weight loss in GLIM.

#### 2.4.2. Low BMI

In LASA, body height (m) was measured to the nearest 0.001 m using a stadiometer. The measured height was adjusted by subtracting 2.7 cm if the participant wore shoes [24]. When no valid height measurement could be obtained due to the recorded particularities such as “not able to stand”, “too much hair”, or the height was missing, one of the following imputation methods was applied: (a) a valid, most recent previous measurement of height was used (14.2% of *n* = 1138 sample); or (b) a self-reported height was used (0.1% of *n* = 1138 sample). Body weight (kg) was measured to the nearest 0.1 kg using a calibrated scale (Seca, model 100; Lameris, Utrecht, The Netherlands). Measured body weight was adjusted by subtracting 1 kg if the participant wore clothes, shoes, or a corset [25]. Self-reported body weight was used when no measured weight was available (12.4% of *n* = 1138 sample). In DNFCS, the participants reported their height and weight with an accuracy of 0.1 kg for body weight and 0.1 cm for height. BMI was calculated as the body weight (kg) divided by the height squared (m). Low BMI was defined as <20.0 kg/m^2^ if the age < 70 years or <22 kg/m^2^ if the age ≥ 70 years [18].

#### 2.4.3. Low MUAC

In both LASA and DNFCS, the measurement of MUAC was taken on the left mid-upper arm to the nearest millimeter at the point equidistant between the lateral projection of the acromion process of the scapula and the inferior margin of the olecranon process of the ulna, with the midpoint determined with the arm flexed at a 90-degree angle. The measurement was taken with the arm hanging passively. Due to COVID-19 restrictions during data collection, only in 10.2% of the DNFCS sample was a MUAC measurement obtained. According to the SNAQ^65+^ criteria, low MUAC was defined as <25 cm [26]. In case the MUAC was missing (12.6% in LASA *n* = 1138 sample and 89.8% in DNFCS *n* = 607 sample), low MUAC was replaced by low BMI.

#### 2.4.4. Decreased Muscle Mass

Calf circumference was used as an indicator of skeletal muscle mass [18]. In LASA only, calf circumference was measured to the nearest 0.001 m (m) on the left leg with the respondent standing straight, the body weight equally distributed on both feet, and at the level of the widest calf circumference. Decreased muscle mass was defined using the cut-off points proposed by Barbosa-Silva et al. [23]: ≤34 cm for men and ≤33 cm for women.

#### 2.4.5. Reduced Food Intake

In LASA only, reduced food intake was determined using the following question: “Have you eaten less than usual in the past 6 months due to loss of appetite, digestion problems, or chewing/swallowing problems?”. Reduced food intake was assumed to be present when answering “less than usual” or “a lot less than usual”.

#### 2.4.6. Poor Appetite

In LASA, one item from the Dutch translation of the Centre for Epidemiologic Studies Depression scale (CES-D) [27] was used to obtain information on appetite: “In the past week, I did not feel like eating, my appetite was poor”. Those that answered “some of the time”, “often”, “most of the time”, or “always” were considered to have a poor appetite [26]. In DNFCS, poor appetite was assessed using the question “In the past week, did you have a reduced appetite?” (yes/no). Those answering "yes" were considered to have a poor appetite. 

#### 2.4.7. Mobility Limitations

In LASA and DNFCS, mobility limitations were assessed by asking “Can you walk up and down a staircase of 15 steps without resting?”. Mobility limitations were based on the response options “with some/much difficulty”, “only with help”, and “no, I cannot” in LASA, and “no” and “wheelchair user” in DNFCS. If the answer was missing for this question, or when the participant indicated not to use the stairs anymore, the following question was used to determine mobility limitations, “Can you walk outside for 5 min without a pause?”, with the same response options.

#### 2.4.8. Inflammation

In LASA only, the presence (yes/no) of chronic non-specific lung disease (CNSLD = chronic obstructive pulmonary disease or asthma) or rheumatoid arthritis in the current and/or previous examination wave, or cancer (excluding skin cancer) in the current examination wave, was used as an indicator of inflammation-related disease [18,28,29]. The self-reported data for these chronic diseases was previously shown to be adequate [30].

### 2.5. Other Descriptive Variables

Age was categorized into: <70 years, 70–74 years, 75–79 years, 80–84 years, and ≥85 years (LASA sample only). A region variable indicated the three regions in which the LASA and DNFCS respondents were living: in the western part of The Netherlands, in the northeast, and in the south. Education was obtained as the highest education level obtained and categorized into three groups: low (elementary school or less), medium (lower vocational or general intermediate education), and high (intermediate vocational education, general secondary school, higher vocational education, college or university). The living situation was dichotomized into “alone” or "with others”. In the LASA sample, self-perceived health was assessed with the question “How do you rate your health in general?" and was dichotomized into (very) good (answer: excellent or good) and less than good (answer: fair, sometimes good/sometimes poor, or poor). In LASA, two questions were also asked to assess the use of formal home care, including personal and household care: “Do you receive help with personal care, for example washing, bathing, dressing?” and “Do you receive help with domestic activities, for example cleaning, shopping, cooking?” The use of any formal home care was dichotomized into yes/no.

### 2.6. Statistical Analysis

Descriptive analyses of characteristics were performed for the study population. Normally distributed continuous variables were presented as the mean with standard deviation (SD) and non-normally distributed variables as the median with an interquartile range (IQR). Normality and linearity were checked by using histograms and scatter plots. Ordinal or categorical variables were presented as absolute numbers and the percentage of total (%). Potential differences in the sample characteristics between the LASA (*n* = 1138) and DNFCS (*n* = 607) were tested for those aged 65–79 years only, using the chi-squared test for the categorical data and the independent sample t-test for the continuous data. As the LASA sample and the GLIM criteria were available for a smaller group than the SNAQ^65+^ criteria, potential differences in the LASA sample characteristics were tested between those with (*n* = 700) and without (*n* = 438) the available GLIM criteria.

The percentage of undernourished participants and those at risk according to the SNAQ^65+^ screening were calculated for the LASA sample, the DNFCS sample, and the two samples combined. The percentage of undernourished participants according to the GLIM criteria were calculated for the LASA subsample with the complete data. The prevalence of undernutrition was presented according to the following variables: sex (male/female); age (<70 years, 70–74 years, 75–79 years, 80–84 years, and ≥85 years); region of The Netherlands (in the western part, in the northeast, and in the south); education (low, medium, and high); self-rated health (good vs. poor, LASA only); appetite (good vs. poor); mobility limitations (no vs. yes); living situation (alone vs. with others); receiving formal home care (yes vs. no, LASA only), and BMI: underweight (<20.0 kg/m^2^ if the age <70 years or <22 kg/m^2^ if the age ≥ 70 years), normal and overweight (22–29 kg/m^2^), and obesity (30+ kg/m^2^). Differences between the groups were tested with the chi-squared test for the nominal data, the Mann-Whitney U test for the ordinal data, and the independent sample t-test was used for the continuous variables. Linear-by-linear associations were calculated to obtain insight into the trend in undernutrition prevalence across subgroups when applicable in the SNAQ^65+^ and GLIM samples. All analyses were performed using SPSS version 27.0 (IBM Armonk, NY, USA: IBM Corp; 2020). *p*-values < 0.05 were considered statistically significant. 

## 3. Results

### 3.1. Sample Characteristics

Table 1 shows the characteristics of the LASA sample, DNFCS, these two samples combined, and the LASA subsample for which the complete data on all GLIM criteria were available. Compared to the LASA participants aged <80 years (*n* = 846), the DNFCS (*n* = 607) participants were somewhat younger (mean age 70.6 (SD 3.9) versus 72.1 (SD 4.1) years, *p* < 0.001), higher educated (high education 62.7% versus 56.4%, *p* = 0.001), less likely to report mobility limitations (4.1% versus 30.7%, *p* < 0.001), and less likely to report a poor appetite (3.0% versus 9.1%, *p* < 0.001) (Appendix A). No differences were observed for sex, region of living, living situation, BMI, and involuntary weight loss. 

Several sample characteristics of the LASA subgroup with the complete data on all GLIM criteria (*n* = 700) differed from those with no complete data (*n* = 438). Those with complete data were more likely to be male (51.6% vs. 43.6%, *p* = 0.01), younger (74.9 years (SD 6.6) vs. 76.2 (SD 7.1), *p* = 0.001), higher educated (57.4% vs. 47.9% in highest education category, *p* = 0.001), living in the western part of the country (47.0% vs. 41.1%, *p* = 0.05), and less likely to receive formal home care (11.6% vs. 19.2%, *p* < 0.001), have poor self-rated health (31.7% vs. 39.5%, *p* = 0.007), have a poor appetite (6.9% vs. 18.5%, *p* < 0.001), report weight loss (4.3% vs. 7.8%, *p* = 0.01), or report mobility limitations (33.0% vs. 46.5%, *p* < 0.001) (Appendix A). Overall, these results suggest that those for whom a GLIM diagnosis of undernutrition could be made were younger, healthier, and better functioning.

### 3.2. Prevalence of Undernutrition Based on SNAQ^65+^ Screening

The prevalence of undernutrition based on screening with the SNAQ^65+^ tool was 8.1% in the LASA sample and 9.4% in the DNFCS sample (Table 1). When combining these two samples (*n* = 1745), the prevalence of undernutrition was 8.5% (95% confidence interval 7.3–9.9%). The prevalence of undernutrition was 5.4% in the LASA subsample for which the complete data on all GLIM criteria were available. The prevalence of being at risk of undernutrition was 4.2% in the combined sample and 3.4% in the LASA subsample with all GLIM criteria. 

### 3.3. Prevalence of Undernutrition Based on GLIM Diagnosis

The overall prevalence of undernutrition according to the GLIM diagnosis was 7.1%. The prevalence of at least one phenotypic criterion was 18.0% in the LASA subgroup for which the complete data on the GLIM criteria were available (Table 2). This percentage varied according to the results of the SNAQ^65+^ screening (*p* < 0.001): 86.8% for those who screened positive for undernutrition, 20.8% for those at-risk, and the lowest (13.8%) for those who screened negative (*p*-value for trend < 0.001). The prevalence of at least one etiologic criterion was 38.0%. This percentage also varied across the SNAQ^65+^ screening categories: 55.3%, 58.3%, and 36.2% (*p*-value for trend 0.007). The prevalence of undernutrition based on the SNAQ65+ screening in this LASA subgroup with the complete GLIM criteria was 5.4% and lower in comparison to the GLIM diagnosis prevalence.

According to the GLIM consensus guideline, only those screened positive (*n* = 38) or at-risk (*n* = 24) based on the screening tool (total *n* = 62 or 8.9%) should be subsequently assessed to diagnose undernutrition. When applying this guideline, the prevalence of undernutrition was 3.1% (see also Figure 1). This large reduction in the prevalence rate was mainly due to the fact that (1) an etiological factor was not present, and (2) less older adults experienced ≥5% involuntary weight loss in the past 6 months (GLIM criterion) as compared to ≥4 kg involuntary weight loss in the past 6 months (SNAQ^65+^ criterion).

### 3.4. Prevalence of Undernutrition by Sample Subgroups

To identify the subgroups of community-dwelling older adults with a higher prevalence of undernutrition, the prevalence according to both screening and diagnosis was calculated according to several sample characteristics (Table 3). The prevalence of undernutrition was higher in females compared to males, the oldest adults (SNAQ^65+^ only), those living alone, receiving formal home care (SNAQ^65+^ only), having mobility limitations, and those with a poor self-rated health or poor appetite. While the prevalence was highest (about 40–50%) in those that are underweight, undernutrition was also present in older adults with normal weight or overweight (about 4–5%) and in those with obesity (about 2–6%). 

## 4. Discussion

Based on the recently collected data in two large and representative samples of Dutch community-dwelling men and women aged 65+ years, the prevalence of undernutrition was 8.5% based on the SNAQ^65+^ screening. The prevalence was highest in those 85 years and older (19.3%). In the LASA subgroup of older adults with the available data on all GLIM criteria, a positive diagnosis of undernutrition was made in 7.1%.

The previous estimate of the prevalence of undernutrition in The Netherlands was based on the data collected in 1998/1999 on 1267 LASA participants aged 65 years and older [15]. At that time, a positive SNAQ^65+^ screening was observed in 10.7%. This somewhat higher prevalence more than 20 years ago for the LASA sample was mainly due to a higher percentage of older adults with a low MUAC (5.8%) compared to the current study (1.6%, *n* = 995 with measured MUAC). The lower prevalence of low MUAC over time for the LASA sample may be partly caused by the obesity epidemic in The Netherlands. The prevalence of obesity in those aged 65–74 years increased from 12.0% in 1998 to 16.8% in 2022, and in those aged 75 years and older from 9.1% to 15.4% [31], resulting in fewer persons with a low MUAC. Despite this increase in body weight, the percentage of older adults with a ≥4 kg involuntary weight loss in the past 6 months in the LASA sample remained relatively stable over the past 20 years (5.4% then versus 5.6% currently). Moreover, the prevalence of being at risk of malnutrition also remained relatively stable over time (7.7% then versus 6.3% currently).

The LASA subgroup of older adults for which the data on all GLIM criteria were available indicated a somewhat lower prevalence of undernutrition (5.4%) based on the SNAQ^65+^ screening as compared to the combined sample (8.5%). This lower prevalence can likely be explained by the fact that this subgroup was somewhat younger, healthier, and had less mobility limitations. Therefore, the prevalence of undernutrition based on GLIM in this LASA subgroup (7.1%) is likely to be an underestimation in comparison to its actual prevalence for the total older population in The Netherlands, and thus should be interpreted with care. This study also demonstrates that applying the GLIM criteria for a diagnosis of undernutrition in the community setting is quite challenging as the missing data on one or more GLIM criteria were often present (for 38% of the LASA sample), and even more so in the older, unhealthier, and less functionally able adults for which the risk of undernutrition is likely highest. Similar to the findings of previous studies, the prevalence of undernutrition based on the GLIM was higher than its prevalence based on the screening tool [32,33]. 

A major strength of this study is that the data were derived from two representative population samples of community-dwelling older adults. Furthermore, the prevalence data were based on screening as well as applying the most recent GLIM consensus definition for the diagnosis of undernutrition. Another strength is that most of the data were collected in the older person’s home, and that information from the telephone interviews in those who were too frail to allow a home visit was also included, allowing more frail older adults to participate.

Some limitations of our study should also be discussed. For many DNFCS participants, a measurement of the MUAC was missing due to COVID-19 restrictions. Therefore, low MUAC was imputed via low BMI based on the self-reported height and weight in order to obtain the SNAQ^65+^ results. This imputation may have somewhat influenced the observed prevalence rates. In a posthoc analysis using only the participants with a measured MUAC were included (*n* = 1057, mean age 75.1 years (SD 6.6)), and the prevalence of undernutrition was somewhat attenuated (6.8%). However, previous research has shown a high correlation (0.83–0.87) between MUAC and BMI [34,35], and the agreement between low MUAC and low BMI in the current study based on the objective measurements was high (85.5%, *n* = 995). It should be acknowledged, however, that the self-reported measures of body weight may be biased. Older adults who are underweight may overestimate their body weight [36], indicating that using self-reported body weight can lead to an underestimation of undernutrition prevalence. However, since the mean BMI and the percentage of adults with a low BMI was similar between the LASA sample and the DNFCS sample for those <80 years of age, this expected bias seems limited. It should be acknowledged that the SNAQ65+ at-risk criterion’s “mobility limitations” was assessed differently between the LASA and the DNFCS sample, causing different prevalences of the “at risk” group. Therefore, the overall prevalence of being at risk (4.2% in the two samples combined) should be interpreted with care. In this study, the GLIM criterion’s “disease burden/inflammatory condition” was based on the presence of CNSLD, rheumatoid arthritis, and cancer. Unfortunately, no information on inflammatory status using a blood marker was available. Finally, it should be acknowledged that older adults not able to read or who are not familiar with the Dutch language are underrepresented, but the potential effect on the prevalence rates is unknown.

The observed prevalence of undernutrition in Dutch community-dwelling older adults indicates that more effort is needed to recognize and treat undernutrition in the community and primary care setting. Effective screening is needed for an early recognition of undernutrition, and most urgently so for the oldest and for those reporting a poor appetite as these subgroups had the highest prevalence. Trained healthcare professionals as well as social care professionals in primary and community care could be involved in this screening process. To avoid delayed treatment, a positive screening should be followed by immediate referral to a dietician for a thorough assessment of nutrition status and personalized nutritional care when necessary. Moreover, risk factors that potentially contributed to the development of undernutrition, including problems swallowing, depression, and poor oral health, should be identified and properly treated by experts (e.g., speech therapist, psychiatrist, or dentist) in order to eliminate or reduce their negative impact on the nutritional status of older adults.

In conclusion, this study shows that 8.5% of the population of Dutch community-dwelling adults aged 65 years and older is currently undernourished based on the SNAQ^65+^ screening and 7.1% based on the GLIM criteria. The prevalence was higher in females, the oldest adults, those living alone, receiving formal home care, and having mobility limitations, and also in those with poor self-rated health or poor appetite. Attention for early recognition and early treatment in the primary care setting is needed, especially among the most vulnerable groups, to support active and healthy aging in this growing segment of the community.

## Figures and Tables

**Figure 1 nutrients-15-03917-f001:**
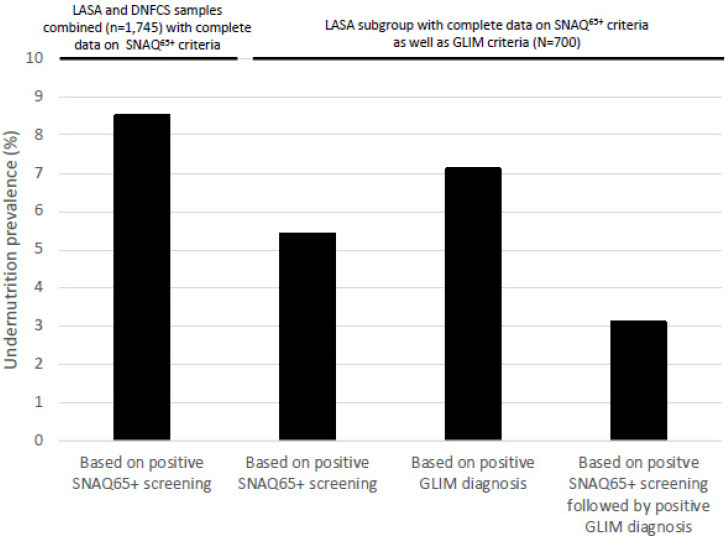
The prevalence of undernutrition in Dutch community-dwelling adults aged 65+ years according to SNAQ65+ screening and GLIM diagnosis. LASA = Longitudinal Aging Study Amsterdam; DNFCS = The Dutch National Food Consumption Survey; GLIM = Global Leadership Initiative on Malnutrition; SNAQ65+ = Short Nutritional Assessment Ques-tionnaire 65+.

**Table 1 nutrients-15-03917-t001:** Characteristics of the study samples of community-dwelling Dutch men and women aged 65+ years and the prevalence of undernutrition according to the SNAQ^65+^ screening.

	UNDERNUTRITION SCREENING SAMPLE (SNAQ^65+^)						UNDERNUTRITION DIAGNOSIS SAMPLE (GLIM)	
	LASA		DNFCS		LASA and DNFCS Combined		LASA SUBGROUP	
	*n*	(N = 1138)	*n*	(N = 607)	*n*	(N = 1745)	*n*	(N = 700)
Sex female, n(%)	1138	586 (51.5)	607	296 (48.8)	1745	882 (50.5)	700	339 (48.4)
Age (years), mean ± SD	1138	75.4 ± 6.9	607	70.7 ± 3.9	1745	73.7 ± 6.4	700	74.9 ± 6.6
Age group, n(%)	1138		607		1745		700	
65–69 years		303 (26.6)		276 (45.5)		579 (33.2)		202 (28.9)
70–74 years		303 (26.6)		213 (35.1)		516 (29.6)		193 (27.6)
75–79 years		240 (22.1)		118 (19.4)		358 (20.5)		144 (20.6)
80–84 years		173 (15.2)		NA		173 (9.9)		96 (13.7)
≥85 years		119 (10.5)		NA		119 (6.8)		65 (9.3)
BMI (kg/m^2^), mean ±SD	1131	27.1 ± 4.7	607	26.9 ± 4.9	1738	27.0 ± 4.8	700	26.7 ± 7
BMI, n (%)	1131		607		1738		700	
Underweight		94 (8.3)		42 (6.9)		136 (7.8)		56 (8.0)
Normal and overweight		796 (70.4)		443 (73.0)		1239 (71.3)		510 (72.9)
Obese		241 (21.3)		122 (20.1)		363 (20.9)		134 (19.1)
Region of The Netherlands, n (%)	1138		607		1745		700	
West		509 (44.7)		282 (46.5)		791 (45.3)		329 (47.0)
Northeast		376 (33.0)		174 (28.7)		550 (31.5)		213 (30.4)
South		253 (22.2)		151 (24.9)		404 (23.2)		158 (22.6)
Education level, n (%)	1138		601		1739		700	
Low		139 (12.2)		27 (4.5)		166 (9.5)		75 (10.7)
Medium		387 (34.0)		197 (32.8)		584 (33.6)		223 (31.9)
High		612 (53.8)		377 (62.7)		989 (56.9)		402 (57.4)
Living situation, n (%)	1125		607		1732		695	
Alone		358 (31.8)		166 (27.3)		524 (30.3)		213 (30.6)
With others		767 (68.2)		441 (72.7)		1208 (69.7)		482 (69.4)
Receiving formal home care, n (%)	1138						700	
Yes		165 (14.5)		NA		NA		81 (11.6)
No		973 (85.5)						619 (88.4)
Self-rated health, n (%)	1138						700	
Good		743 (65.3)		NA		NA		478 (68.3)
Poor		395 (34.7)						222 (31.7)
**Individual SNAQ**								
≥4 kg involuntary weight loss, n (%)	1138	64 (5.6)	607	25 (4.1)	1745	89 (5.1)	700	30 (4.3)
MUAC < 25 cm, n (%) *	1138	33 (2.9)	607	34 (5.6)	1745	67 (3.8)	700	11 (1.6)
Appetite last week, n (%)	1138		607		1745		700	
Good		1009 (88.7)		589 (97.0)		1598 (91.6)		652 (93.1)
Poor		129 (11.3)		18 (3.0)		147 (8.4)		48 (6.9)
Mobility limitations, n (%)	1137		607		1744		700	
No		703 (61.8)		582 (95.9)		1285 (73.7)		469 (67.0)
Yes		434 (38.2)		25 (4.1)		459 (26.3)		231 (33.0)
**Screening result SNAQ^65+^**								
Undernutrition, n (%)	1138	92 (8.1)	607	57 (9.4)	1745	149 (8.5)	700	38 (5.4)
At-risk, n (%)	1138	72 (6.3)	607	1 (0.2)	1745	73 (4.2)	700	24 (3.4)
None, n (%)	1138	974 (85.6)	607	549 (90.4)	1745	1523 (87.3)	700	638 (91.1)

LASA = Longitudinal Aging Study Amsterdam; DNFCS = The Dutch National Food Consumption Survey; SNAQ^65+^ = Short Nutritional Assessment Questionnaire 65+; GLIM = Global Leadership Initiative on Malnutrition; MUAC = mid-upper arm circumference; SD = standard deviation; BMI = Body Mass Index; NA = not available. * with BMI imputation, when MUAC was missing. The N indicates the sample size of the total groups, the n the number of people for which a variable was available.

**Table 2 nutrients-15-03917-t002:** Prevalence of undernutrition diagnosis according to the GLIM criteria in the LASA subgroup of community-dwelling Dutch men and women aged 65+ years (*n* = 700), also stratified by the SNAQ^65+^ screening result.

	ALL	SNAQ^65+^ Screening Result		
		Undernutrition	At Risk	None
N	700	38	24	638
**Phenotypic criteria, n (%)**				
Unintentional weight loss ≥5% in 6 months	22 (3.1)	19 (50.0)	1 (4.2)	2 (0.3)
*Severe*	5 (0.7)	4 (10.5)	0 (0.0)	1 (0.2)
Low BMI				
*Severe*	56 (8.0)	15 (39.5)	1 (4.2)	40 (6.3)
	13 (1.9)	8 (21.1)	0 (0.0)	5 (0.8)
Reduced muscle mass	74 (10.6)	10 (26.3)	5 (20.8)	59 (9.2)
*Severe*	16 (2.3)	4 (10.5)	1 (4.2)	11 (1.7)
At least one phenotypic criterion	126 (18.0)	33 (86.8)	5 (20.8)	88 (13.8)
**Etiologic criteria, n (%)**				
Reduced intake	69 (9.9)	2 (5.3)	2 (8.3)	65 (10.2)
Inflammation	227 (32.4)	20 (52.6)	13 (54.2)	194 (30.4)
*CNSLD*	85 (12.1)	7 (18.4)	5 (20.8)	73 (11.4)
*Rheumatoid arthritis*	87 (12.4)	6 (15.8)	6 (25.0)	75 (11.8)
*Non-skin cancer*	92 (13.1)	9 (23.7)	4 (16.7)	79 (12.4)
At least one etiologic criterion	266 (38.0)	21 (55.3)	14 (58.3)	231 (36.2)
**Diagnosis based on GLIM**				
Undernutrition	50 (7.1)	19 (50.0)	3 (12.5)	28 (4.4)
*Moderate undernutrition*	24 (3.4)	9 (23.7)	2 (8.3)	13 (2.0)
*Severe undernutrition*	26 (3.7)	10 (26.3)	1 (4.2)	15 (2.4)
No undernutrition	650 (92.9)	19 (50.0)	21 (87.5)	610 (95.6)

SNAQ^65+^ = Short Nutritional Assessment Questionnaire 65+; GLIM = Global Leadership Initiative on Malnutrition; CNSLD = chronic non-specific lung disease; BMI = Body Mass Index. Italics indicate that these group are subgroups of the parameter just above

**Table 3 nutrients-15-03917-t003:** Prevalence of undernutrition in community-dwelling Dutch older men and women aged 65+ years based on the SNAQ^65+^ screening and based on the GLIM diagnosis, according to several sample characteristics.

	UNDERNUTRITION SCREENING SAMPLE (SNAQ^65+^)		UNDERNUTRITION DIAGNOSIS SAMPLE (GLIM)	
	LASA and DNFCS Combined	*p*-Value *	LASA Subgroup	*p*-Value *
N	1745		700	
Sex, n (%)		0.001		0.16
male	55 (6.4)		21 (5.8)	
female	94 (10.7)		29 (8.6)	
Age, n (%)		<0.001		0.46
65–69 years	28 (4.8)		11 (5.4)	
70–74 years	48 (9.3)		14 (7.3)	
75–79 years	35 (9.8)		15 (10.4)	
80–84 years **	15 (8.7)		4 (4.2)	
≥85 years **	23 (19.3)		6 (9.2)	
BMI, n (%)		<0.001		<0.001
Underweight	72 (52.9)		23 (41.1)	
Normal and overweight	51 (4.1)		24 (4.7)	
Obese	22 (6.1)		3 (2.2)	
Appetite, n (%)		<0.001		<0.001
Good	116 (7.3)		39 (6.0)	
Poor	33 (22.4)		11 (22.9)	
Mobility limitations, n (%)		0.007		0.008
No	96 (7.5)		25 (5.3)	
Yes	53 (11.5)		25 (10.8)	
Region, n (%)		0.39		0.29
West	74 (9.4)		28 (8.5)	
Northeast	42 (7.6)		12 (5.6)	
South	33 (8.2)		10 (6.3)	
Education, n (%)		0.24		0.07
Low	17 (10.2)		10 (13.3)	
Medium	53 (9.1)		15 (6.7)	
High	78 (7.9)		25 (6.2)	
Living situation, n (%)		<0.001		0.07
Alone	71 (13.5)		23 (10.8)	
With others	76 (6.3)		25 (5.2)	
Receiving formal home care, n(%) **		<0.001		0.72
Yes	26 (15.8)		5 (6.2)	
No	66 (6.8)		45 (7.3)	
Self-rated health, n (%) **		<0.001		0.05
Good	44 (5.9)		28 (5.9)	
Poor	8 (12.2)		22 (9.9)	

* Test for trend across categories, ** LASA sample only LASA = Longitudinal Aging Study Amsterdam; DNFCS = The Dutch National Food Consumption Survey; GLIM = Global Leadership Initiative on Malnutrition; SNAQ^65+^ = Short Nutritional Assessment Questionnaire 65+; BMI = Body Mass Index.

## Data Availability

The data presented in this study are available on request from the corresponding author. LASA data and DNFCS data are available for research. More information can be found on the LASA website: www.lasa-vu.nl and the DNFCS website: www.wateetnederland.nl.

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
