# Peer review of "Prevalence of Undernutrition in Community-Dwelling Older Adults in The Netherlands: Application of the SNAQ65+ Screening Tool and GLIM Consensus Criteria"

_nutrients, 2023, doi:10.3390/nu15183917_

Round 1

Reviewer 1 Report

This manuscript aims to determine the current prevalence of undernutrition in Dutch community-dwelling older adults using the SNAQ65+ screening tool as well as the Global Leadership Initiative on Malnutrition (GLIM) criteria. I have some concerns about the objective and conclusion.

1. Why did the authors use two methods to determine the current prevalence of undernutrition in Dutch community-dwelling older adultsThis needs to be explained in detail in the preface.

2. What is the purpose of presenting the prevalence in Dutch community-dwelling older adults through SNAQ65+ and GLIM, respectively? What is the value of this result? Which result should the reader refer to?

3. If the authors want to compare the results of SNAQ65+ and GLIM, they should do so for the same sample when analyzing the data. Whether analyzing the LASA or DNFCS samples, or the combined samples.

Author Response

Reviewer 1:

This manuscript aims to determine the current prevalence of undernutrition in Dutch community-dwelling older adults using the SNAQ65+ screening tool as well as the Global Leadership Initiative on Malnutrition (GLIM) criteria. I have some concerns about the objective and conclusion.

  1. Why did the authors use two methods to determine the current prevalence of undernutrition in Dutch community-dwelling older adults? This needs to be explained in detail in the preface.

As requested, we now explain better in the introduction section why the SNAQ65+ criteria as well as the GLIM criteria were used to assess the prevalence of undernutrition. We have added the following sentence on line 59: “This tool has been developed for the Netherlands and is the most frequently used tool for older adults in the community setting in this country.”  And to line 71: “…….., recent prevalence data using the same screening tool on Dutch…..”. And added this sentence (line 73): “Using consistent methodology allows the potential comparison of prevalences over time in the Netherlands.” We have also better motivated the use of the GLIM criteria in the introduction section by adding (line 85): “A prevalence rate according to these newly developed and global criteria for the assessment of undernutrition importantly allows comparisons of prevalences between countries and between settings.” And finally, we have added the words “for diagnosis” to our aim in line 90, to better highlight the difference (screening versus diagnosis) between the two methods. The word “diagnosis” was also added to the abstract (line 33) to better highlight the difference between the two tools.

  1. What is the purpose of presenting the prevalence in Dutch community-dwelling older adults through SNAQ65+ and GLIM, respectively? What is the value of this result? Which result should the reader refer to?

The response to this comment has been included in the response to comment 1 (please see above).

  1. If the authors want to compare the results of SNAQ65+ and GLIM, they should do so for the same sample when analyzing the data. Whether analyzing the LASA or DNFCS samples, or the combined samples.

We like to refer the reviewer to table 2. This table shows in the same sample (n=700) the prevalence results of both the SNAQ65+ and the GLIM.  These results are also presented in Figure 1. However, in order to better highlight the direct comparison between the two tools, the following sentence has been added to the results section (line 346): “The prevalence of undernutrition based on the SNAQ65+ screening in this LASA subgroup with complete GLIM criteria was 5.4% and lower in comparison to the GLIM diagnosis prevalence.

Reviewer 2 Report

Presently, the manuscript investigate the prevalence of undernutrition by the positive score on the SNAQ65+ and GLIM criteria, and their combination. The results indicated that 1/12 or 1/14 of the community-dwelling aduls aged 65 years and older is undernourished based on the above-mentioned two methods. There are some issued should be improved.

1. The abstract can be written more succinctly.

2. Why the authors chose the methods of SNAQ65+ and GLIM criteria to investigate the prevalence of undernutrition? If there have any other evaluation methodology?

3. The clarity of Figure 1 needs improvement.

4. As the authors metioned, the data of MUAC missed due to COVID-19, how to resolve this problem during the analysis? Beside, any other data were incomplete?

5. What the reasons that lead to the high prevalence of undernutrition should be discussed, as well as the response strategy.   

Author Response

Reviewer 2:

Presently, the manuscript investigate the prevalence of undernutrition by the positive score on the SNAQ65+ and GLIM criteria, and their combination. The results indicated that 1/12 or 1/14 of the community-dwelling aduls aged 65 years and older is undernourished based on the above-mentioned two methods. There are some issued should be improved.

  1. The abstract can be written more succinctly.

We appreciate this rather general comment, but are in doubt what changes the reviewer would like us to make as no specific comments were provided. In our opinion, the abstract is an accurate and complete representation of our study, nor did reviewer 1 comment on the contents of the abstract. However, if the editor would like us to make some changes, we are more than happy to do so based on your specific suggestions.

  1. Why the authors chose the methods of SNAQ65+ and GLIM criteria to investigate the prevalence of undernutrition? If there have any other evaluation methodology?

The question raised by reviewer 2 is very similar to comment 1 and comment 2 of reviewer 2. Please find our response there.

  1. The clarity of Figure 1 needs improvement.

The group descriptions as well as the x-axis titles of figure 1 have been extended in order to better clarify this figure.

  1. As the authors metioned, the data of MUAC missed due to COVID-19, how to resolve this problem during the analysis? Beside, any other data were incomplete?

In the methods section, we already described that the measurement of MUAC could not take place in part of the study population due to COVID-19 restrictions (lines 203-207). We kindly refer the reviewer to this section, as it also describes how we dealt with these missing data in the analyses (lines 206-207). Regarding other missing data, in the DNFCS sample no other data were missing. In the LASA study,  similar to any other large-scale prospective cohort study, some study participants had missing data. These were described in detail in lines 124-128 for the large LASA sample for which SNAQ65+ criteria were available, and in lines 129-132 for the LASA subgroup for which GLIM criteria were available. As these missing data may affect the representativeness of our LASA study samples, we have performed statistical analyses in order to compare study participants who were in- and excluded for the current analysis (see method section lines 269-276). The results of these comparisons are described in detail in the results section (lines 301-307 and lines 308-319).

  1. What the reasons that lead to the high prevalence of undernutrition should be discussed, as well as the response strategy. 

As requested by the reviewer, we have added a paragraph (lines 461-475) on the strategy to reduce the prevalence of undernutrition in older adults, and in which we also highlight the importance of identifying risk factors. “The observed prevalence of undernutrition in Dutch community-dwelling older adults indicates that more effort is needed to recognize and treat undernutrition in the community and primary care setting. Effective screening is needed for an early recognition of undernutrition, and most urgently so in the oldest old and those reporting a poor appetite as these subgroups had the highest prevalence. Trained healthcare professionals as well as social care professionals in primary and community care could be involved in this screening process. To avoid delayed treatment, a positive screening should be followed by immediate referral to a dietician for a thorough assessment of nutrition status and personalized nutritional care when necessary. Moreover, risk factors that potentially contributed to the development of undernutrition, including problems swallowing, depression and poor oral health, should be identified and properly treated by experts (e.g., speech therapist, psychiatrist or dentist) in order to eliminate or reduce their negative impact on the nutritional status of older adults.” 

Round 2

Reviewer 1 Report

I would like to thank the authors for their reply. I have no further comments.